# Liver injury in hospitalized patients with COVID-19: An International observational cohort study

Bharath Kumar Tirupakuzhi Vijayaraghavan[1,2]*, Saptarshi Bishnu[3], Joaquin Baruch[2], Barbara Wanjiru Citarella[2], Christiana Kartsonaki[4], Aronrag Meeyai[2], Zubair Mohamed[5], Shinichiro Ohshimo[6], Benjamin Lefèvre[7,8], Abdulrahman Al-Fares[9,10], Jose A. Calvache[11,12], Fabio Silvio Taccone[13], Piero Olliaro[2], Laura Merson[2], Neill K. J. Adhikari[14], the ISARIC Clinical Characterisation Group[¶]

1 Department of Critical Care Medicine, Apollo Main Hospital, Chennai, India and Honorary Senior Fellow, The George Institute for Global Health, New Delhi, India, 2 ISARIC, Pandemic Science Institute, University of Oxford, Oxford, United Kingdom, 3 Department of Hepatology, Apollo Main Hospital, Chennai, India and Department of Gastroenterology and Hepatology, Sharanya Multi-specialty Hospital, Burdwan, India, 4 MRC Population Health Research Unit, Clinical Trials Service Unit and Epidemiological Studies Unit, Nuffield Department of Population Health, University of Oxford, Oxford, United Kingdom, 5 Department of Critical Care Medicine, Amrita Institute of Medical Sciences, Kochi, Kerala, India, 6 Department of Emergency and Critical Care Medicine, Graduate School of Biomedical and Health Sciences, Hiroshima University, Hiroshima, Japan, 7 Université de Lorraine, CHRU-Nancy, Service des Maladies Infectieuses et Tropicales, Nancy, France, 8 Université de Lorraine, APEMAC, Nancy, France, 9 Department of Anaesthesia, Critical Care Medicine and Pain Medicine, Al-Amiri Hospital, Ministry of Health, Kuwait, Kuwait, 10 Kuwait Extracorporeal Life Support program, Al-Amiri Hospital, Ministry of Health, Kuwait, Kuwait, 11 Departamento de Anestesiologia, Universidad del Cauca, Popayan, Colombia, 12 Department of Anaesthesiology, Erasmus University Medical Center, Rotterdam, The Netherlands, 13 Department of Intensive Care Medicine, Erasme Hospital, Universite Libre de Bruxelles, Brussels, Belgium, 14 Interdepartmental Division of Critical Care Medicine, University of Toronto and Department of Critical Care Medicine, Sunnybrook Health Sciences Centre, Toronto, Canada

¶ Membership of the ISARIC Clinical Characterisation Group is provided in the Acknowledgments. The lead author for this group is Laura Merson.
* bharath@icuconsultants.com

**Data Availability Statement:** The minimal dataset underlying the results are available within the manuscript and supplementary files. Any additional data that relate to this analysis are highly detailed

## Abstract

### Background

Using a large dataset, we evaluated prevalence and severity of alterations in liver enzymes in COVID-19 and association with patient-centred outcomes.

### Methods

We included hospitalized patients with confirmed or suspected SARS-CoV-2 infection from the International Severe Acute Respiratory and emerging Infection Consortium (ISARIC) database. Key exposure was baseline liver enzymes (AST, ALT, bilirubin). Patients were assigned Liver Injury Classification score based on 3 components of enzymes at admission: Normal; Stage I) Liver injury: any component between 1-3x upper limit of normal (ULN); Stage II) Severe liver injury: any component ≥3x ULN. Outcomes were hospital mortality, utilization of selected resources, complications, and durations of hospital and ICU stay.

clinical data on individuals hospitalised with COVID-19. Due to the sensitive nature of these data and the associated privacy concerns, they are available via a governed data access mechanism following review of a data access committee. Data can be requested via the IDDO COVID-19 Data Sharing Platform (http://www.iddo.org/covid-19). The email is: dataaccess@iddo.org. The Data Access Application, Terms of Access and details of the Data Access Committee are available on the website. Briefly, the requirements for access are a request from a qualified researcher working with a legal entity who have a health and/or research remit; a scientifically valid reason for data access which adheres to appropriate ethical principles. The full terms are at https://www.iddo.org/document/covid-19-data-access-guidelines. A small subset of sites who contributed data to this analysis have not agreed to pooled data sharing as above. In the case of requiring access to these data, please contact the corresponding author in the first instance who will look to facilitate access.

**Funding:** "This work was made possible by the UK Foreign, Commonwealth and Development Office and Wellcome [215091/Z/18/Z, 222410/Z/21/Z, 225288/Z/22/Z]; and the Bill & Melinda Gates Foundation [OPP1209135]. The funders had no role in the design, analysis, manuscript preparation or decision to submit for publication."

**Competing interests:** The authors have declared that no competing interests exist

Analyses used logistic regression with associations expressed as adjusted odds ratios (OR) with 95% confidence intervals (CI).

## Results

Of 17,531 included patients, 46.2% (8099) and 8.2% (1430) of patients had stage 1 and 2 liver injury respectively. Compared to normal, stages 1 and 2 were associated with higher odds of mortality (OR 1.53 [1.37–1.71]; OR 2.50 [2.10–2.96]), ICU admission (OR 1.63 [1.48–1.79]; OR 1.90 [1.62–2.23]), and invasive mechanical ventilation (OR 1.43 [1.27–1.70]; OR 1.95 (1.55–2.45). Stages 1 and 2 were also associated with higher odds of developing sepsis (OR 1.38 [1.27–1.50]; OR 1.46 [1.25–1.70]), acute kidney injury (OR 1.13 [1.00–1.27]; OR 1.59 [1.32–1.91]), and acute respiratory distress syndrome (OR 1.38 [1.22–1.55]; OR 1.80 [1.49–2.17]).

## Conclusions

Liver enzyme abnormalities are common among COVID-19 patients and associated with worse outcomes.

## Introduction

Coronavirus Disease 2019 (COVID-19) caused by the Severe Acute Respiratory Syndrome Coronavirus 2 (SARS-CoV-2) has thus far resulted in over 6.9 million deaths globally [1] and continues to contribute to substantial morbidity and mortality. Our understanding of COVID-19 has considerably evolved from the time the first case was reported in December 2019, and while pulmonary manifestations predominate, multi-organ involvement is well-described [2].

As a component of multi-organ involvement, liver injury, defined by elevated liver enzymes (alanine aminotransferase [ALT], aspartate aminotransferase [AST], and serum bilirubin), has been reported in 15–65% of patients [3–7]. Abnormalities in liver enzymes have been associated with severe COVID-19 and an increased risk of death [4, 6, 8, 9]. Multiple mechanisms may contribute to liver injury in COVID-19, including direct viral toxicity, endothelial damage and immune dysfunction [2]. Drugs used to treat patients with COVID-19, such as remdesivir, may also contribute to liver injury [10]. However, current information on the extent and severity of liver enzyme derangements and their implications for clinical practice come predominantly from small single-centre studies.

In January 2020, the International Severe Acute Respiratory and emerging Infection Consortium (ISARIC) [11], in partnership with the World Health Organization (WHO), activated the ISARIC-WHO Clinical Characterisation Protocol and case report form (CRF) to collect data on demographics, illness severity, treatment strategies and outcomes for hospitalized patients with COVID-19 [12, 13]. ISARIC hosts data for the largest world-wide cohort of hospitalized COVID-19 patients.

Using the ISARIC dataset, we evaluated the prevalence and severity of derangements in liver enzymes among patients admitted to hospital with COVID-19 and the association between liver enzymes measured during the first 24 hours and patient-centred hospital outcomes.

## Methods

### Study design and ethics

The ISARIC-WHO Clinical Characterization Protocol for Severe Emerging Infections provided the framework for prospective observational data collection on hospitalised patients with COVID-19. The protocol, CRFs, consent forms, and study information are available online [14]. These CRFs were developed to standardise clinical data collection on patients admitted with suspected or confirmed COVID-19, with clinical data on more than 900,000 individuals hospitalised with confirmed COVID-19 infection across 64 countries stored in a central database (as of June 2022). The CRFs collect data on demographics, pre-existing comorbidities and risk factors, signs and symptoms during the acute phase, supportive care and treatments received during hospitalisation, and outcomes [14].

This observational study required no change to clinical management. The ISARIC-WHO Clinical Characterisation Protocol was approved by the World Health Organization Ethics Review Committee (RPC571 and RPC572 on 25 April 2013). Institutional approval was additionally obtained by participating sites including the South Central Oxford C Research Ethics Committee in England (Ref 13/SC/0149) and the Scotland A Research Ethics Committee (Ref 20/SS/0028) for the United Kingdom, representing the majority of the data. Other institutional and national approvals were obtained by participating sites as per local requirements. Regionally appropriate decisions regarding a waiver or requirement of patient consent were made by each committee and implemented at the sites. A statistical analysis plan was developed *a priori* for this study and reviewed by the ISARIC Clinical-Analytic Team as well as by partner sites and collaborators (S1 File). This study is being reported as per the Strengthening the Reporting of Observational Studies in Epidemiology (STROBE) guidelines [15] (S3 Table).

### Study population

We included all individuals in the ISARIC database with laboratory confirmed or suspected SARS-CoV-2 infection admitted to hospital from 30/01/2020 to 21/09/2021 for the primary analysis. For the sensitivity analysis, we included only patients with laboratory-confirmed SARS-CoV-2 infection. We excluded patients for whom information on liver enzyme tests or clinical outcomes were not available.

### Variables and definitions

Serum bilirubin, ALT, and AST measured at or within 24 hours of hospital admission were considered as the liver enzymes for this analysis, based on availability in the ISARIC database. For the purposes of this analysis, upper limits of normal (ULN) for serum bilirubin, ALT and AST were taken as 1 mg/dL, 40 U/L and 40 U/L respectively. In the absence of an established scale for liver injury based in these laboratory tests, we adapted our criteria from previous studies that have used similar approaches [8, 9, 16–18]. Each patient was assigned a Liver Injury Classification (LIC) score at baseline based on the 3 components of LFT on admission: stage 0) Normal: all 3 components ≤ULN; Stage I) Liver injury: any 1 component between 1-3x ULN; Stage II) Severe liver injury: any 1 component ≥3x ULN.

**Main exposures and outcomes.** For the primary research question, the main exposure was baseline liver enzymes (defined as above) and the primary outcome was hospital mortality. Secondary outcomes included admission to an intensive care unit (ICU); receipt of oxygen therapy, non-invasive ventilation (NIV) or invasive ventilation, inotropes/vasopressors, and renal replacement therapy; and the durations of hospital and ICU stay. We also performed additional analyses analysis examining the association between baseline liver enzymes and

specific complications not present at admission and developing in hospital, including acute respiratory distress syndrome (ARDS), hemodynamic complications, acute kidney injury, sepsis, and hematological and neurological complications. Definitions for these complications are available from the ISARIC CRF completion guide [19]. Other baseline exposures were included based on biological relevance: comorbidities, age (in ten-year bands), sex, and depending on the model, ICU admission and receipt of oxygen.

## Statistical analysis

Categorical variables were summarized as counts and percentages and continuous variables as mean ± standard deviation (SD) or median (first quartile [Q3], third quartile [Q3]), depending on distribution. The prevalence of liver enzyme derangements at baseline was estimated using the LIC classification described. The cumulative probability of a patient remaining in hospital or ICU (i.e., length of stay) was plotted graphically, stratified by discharge vital status and age.

We used logistic regression to determine the association between exposure variables and outcomes, expressed as odds ratios (ORs) with 95% confidence intervals (CIs). Dichotomous outcomes were analysed using a binomial distribution and a logit link. As mentioned above, covariates were selected based on the biological relevance. For the remaining covariates, those with P <0.20 in univariable analysis were retained for the multivariable analysis, and backwards elimination was used for model selection. To account for potential effect modifications, two-way interactions were evaluated between LIC and age categories, ICU admission, and receipt of remdesivir. Missing data were not imputed, and analyses used complete cases. In a sensitivity analysis (primary outcome), we included only laboratory confirmed patients. We used R4.1.2 [R Core Team. R: A language and environment for statistical computing. R Foundation for Statistical Computing, Vienna, Austria. https://www.R-project.org/] for statistical analysis.

**Deviations from original analysis plan.** We originally planned to examine the association between receipt of remdesivir and new liver enzyme elevation, as well as at trends of liver enzymes during the hospital stay. Both these analyses were not possible due to the high degree of missingness for these variables in the dataset.

## Results

From a total of 708,052 patients in the ISARIC database as of 21st September 2021, we included 17,531 patients after eliminating those with missing baseline information on liver enzyme tests (n = 686,122 patients) and outcomes (4399 patients). Baseline characteristics of included patients are presented by stages of LIC classification in Table 1. The mean age was 56.5 (SD 20.3) years, and 60.0% of the patients were male. Hypertension (33.9%) and diabetes (31.3%) were the most common comorbidities. Around 3.0% of the cohort had chronic liver disease. Cough (60.0%) and fever (59.0%) were the most common presenting symptoms of COVID-19.

45.6% of patients had normal liver enzymes, 46.2% had stage 1 liver injury, and 8.2% of the cohort had stage 2 liver injury (Table 1). Patients met criteria for liver injury predominantly through elevations in AST or ALT (S2 Table).

Treatments received during hospital stay are presented by stages of LIC in Table 2. Admission to an ICU for patients with LIC stages 0, 1, and 2 occurred in 1514 (18.9%), 2834 (35.0%) and 573 (40.1%) patients, respectively. Oxygen supplementation was provided in 3838 (48.0%), 5692 (70.3%), and 1086 (75.9%) of patients. 782 patients (9.8%) in stage 0, 1722 (21.3%) in stage 1, and 386 (27.0%) in stage 2 received invasive ventilation. Corticosteroids were administered to 2294 (28.7%, stage 0), 3446 (42.5%, stage 1), and 656 (45.9%, stage 2) patients. The median (IQR) length of hospital stay for stages 0, 1 and 2 were 9 (5–15) days, 8

**Table 1. Baseline characteristics by stages of liver injury in the ISARIC clinical characterisation database (n = 17531).**

| Characteristic | Stage 0 (n = 8002) | Stage 1 (n = 8099) | Stage 2 (n = 1430) |
|---|---|---|---|
| **Age (mean ± SD)** | 56.0 ± 21.5 | 60.8 ± 18.8 | 57.4 ± 19.0 |
| **Sex (n, %)** | | | |
| Male | 3732 (46.6) | 2661 (32.9) | 435 (30.4) |
| Female | 4264 (53.3) | 5421 (66.9) | 992 (69.4) |
| Unknown | 6 (0.1) | 17 (0.2) | 3 (0.2) |
| **Comorbidities** | | | |
| Hypertension (n, %) | | | |
| No | 4524 (56.5) | 4010 (49.5) | 732 (51.2) |
| Yes | 2576 (32.2) | 2890 (35.7) | 472 (33) |
| Yes | 2576 (32.2) | 2890 (35.7) | 472 (33) |
| Unknown | 902 (11.3) | 1199 (14.8) | 226 (15.8) |
| Chronic liver disease (n, %) | | | |
| No | 7736 (96.7) | 7657 (94.5) | 1284 (89.8) |
| Yes | 153 (1.9) | 252 (3.1) | 92 (6.4) |
| Unknown | 113 (1.4) | 190 (2.3) | 54 (3.8) |
| Chronic kidney disease (n, %) | | | |
| No | 6992 (87.4) | 7141 (88.2) | 1209 (84.5) |
| Yes | 781 (9.8) | 619 (7.6) | 143 (10) |
| Unknown | 229 (2.9) | 339 (4.2) | 78 (5.5) |
| Chronic neurological disorder (n, %) | | | |
| No | 7198 (90) | 7259 (89.6) | 1252 (87.6) |
| Yes | 561 (7) | 502 (6.2) | 101 (7.1) |
| Unknown | 243 (3) | 338 (4.2) | 77 (5.4) |
| Chronic pulmonary disease (n, %) | | | |
| No | 6932 (86.6) | 6995 (86.4) | 1217 (85.1) |
| Yes | 858 (10.7) | 794 (9.8) | 137 (9.6) |
| Unknown | 212 (2.6) | 310 (3.8) | 76 (5.3) |
| Malignant neoplasm (n, %) | | | |
| No | 7301 (91.2) | 7298 (90.1) | 1249 (87.3) |
| Yes | 477 (6) | 491 (6.1) | 115 (8) |
| Unknown | 224 (2.8) | 310 (3.8) | 66 (4.6) |
| Obesity (n, %) | | | |
| No | 5731 (71.6) | 5292 (65.3) | 917 (64.1) |
| Yes | 822 (10.3) | 1136 (14) | 200 (14) |
| Unknown | 1449 (18.1) | 1671 (20.6) | 313 (21.9) |
| **Symptoms** | | | |
| Cough (n, %) | | | |
| No | 3354 (41.9) | 2498 (30.8) | 491 (34.3) |
| Yes | 4385 (54.8) | 5281 (65.2) | 838 (58.6) |
| Unknown | 263 (3.3) | 320 (4.0) | 101 (7.1) |
| History of fever (n, %) | | | |
| No | 3470 (43.4) | 2512 (31) | 492 (34.4) |
| Yes | 4222 (52.8) | 5259 (64.9) | 850 (59.4) |
| Unknown | 310 (3.9) | 328 (4.0) | 88 (6.2) |
| lost altered sense of smell (n, %) | | | |
| No | 4986 (62.3) | 4668 (57.6) | 830 (58) |
| Yes | 397 (5) | 446 (5.5) | 63 (4.4) |

*(Continued)*

**Table 1.** (Continued)

| Characteristic | Stage 0 (n = 8002) | Stage 1 (n = 8099) | Stage 2 (n = 1430) |
|---|---|---|---|
| Unknown | 2619 (32.7) | 2985 (36.9) | 537 (37.6) |
| Shortness of breath (n, %) | | | |
| No | 4096 (51.2) | 2666 (32.9) | 428 (29.9) |
| Yes | 3652 (45.6) | 5199 (64.2) | 937 (65.5) |
| Unknown | 254 (3.2) | 234 (2.9) | 65 (4.5) |

(5–16) days, and 9 (4–14) days, respectively. Similarly, the median (IQR) length of ICU stay for stages 0, 1 and 2 were 7 (4–15) days, 8 (5–17) days, and 9 (3–14) days, respectively (S1 and S2 Figs).

The crude risk of death was 14.3% in stage 0, 23.4% in stage 1, and 32.7% in stage 2 (Table 3).

**Table 2. Interventions and treatments during hospital stay among patients in the ISARIC clinical characterisation database with different levels of liver injury (n = 17531).**

| Intervention | Stage 0 (n = 8002) | Stage 1 (n = 8099) | Stage 2 (n = 1430) |
|---|---|---|---|
| ICU admission (n, %) | | | |
| No | 6375 (79.7) | 5139 (63.5) | 823 (57.6) |
| Yes | 1514 (18.9) | 2834 (35.0) | 573 (40.1) |
| Unknown | 113 (1.4) | 126 (1.6) | 34 (2.4) |
| Oxygen therapy (n, %) | | | |
| No | 4123 (51.5) | 2377 (29.3) | 338 (23.6) |
| Yes | 3838 (48.0) | 5692 (70.3) | 1086 (75.9) |
| Unknown | 41 (0.5) | 30 (0.4) | 6 (0.4) |
| Non-invasive ventilation (n, %) | | | |
| No | 7136 (89.2) | 6494 (80.2) | 1149 (80.3) |
| Yes | 712 (8.9) | 1444 (17.8) | 260 (18.2) |
| Unknown | 154 (1.9) | 161 (2.0) | 21 (1.5) |
| Invasive ventilation (n, %) | | | |
| No | 7141 (89.2) | 6308 (77.9) | 1034 (72.3) |
| Yes | 782 (9.8) | 1722 (21.3) | 386 (27) |
| Unknown | 79 (1.0) | 69 (0.9) | 10 (0.7) |
| Inotropes/vasopressors (n, %) | | | |
| No | 7088 (88.6) | 6381 (78.8) | 1064 (74.4) |
| Yes | 701 (8.8) | 1466 (18.1) | 322 (22.5) |
| Unknown | 213 (2.7) | 252 (3.1) | 44 (3.1) |
| Corticosteroids (n, %) | | | |
| No | 5452 (68.1) | 4366 (53.9) | 703 (49.2) |
| Yes | 2294 (28.7) | 3446 (42.5) | 656 (45.9) |
| Unknown | 256 (3.2) | 287 (3.5) | 71 (5.0) |
| Antiviral (n, %) | | | |
| No | 4250 (53.1) | 4897 (60.5) | 964 (67.4) |
| Yes | 1530 (19.1) | 1985 (24.5) | 303 (21.2) |
| Unknown | 2222 (27.8) | 1217 (15) | 163 (11.4) |
| Antibiotics (n, %) | | | |
| No | 1407 (17.6) | 1038 (12.8) | 177 (12.4) |
| Yes | 4163 (52) | 5728 (70.7) | 1079 (75.5) |
| Unknown | 2432 (30.4) | 1333 (16.5) | 174 (12.2) |

**Table 3. Unadjusted risk of death by LIC score in the ISARIC clinical characterisation data base (n = 17531).**

| Outcome | Stage 0 (n = 8002) | Stage 1 (n = 8099) | Stage 2 (n = 1430) |
|---|---|---|---|
| Death (n, %) | 1145 (14.3) | 1893 (23.4) | 467 (32.7) |
| Discharge (n, %) | 6857 (85.7) | 6206 (76.6) | 963 (67.3) |

In multivariable analysis (Table 4), compared to normal, stages 1 and 2 were associated with higher odds of mortality (OR 1.53 [1.37–1.71]; OR 2.50 [2.10–2.96]), ICU admission (OR 1.63 [1.48–1.79]; OR 1.90 [1.62–2.23]) and invasive mechanical ventilation (OR 1.43 [1.27–1.70]; OR 1.95 [1.55–2.45]).

Associations of LIC with complications are shown in Table 5. When comparing to stage 0, stage 1 and 2 of LIC were associated with a higher odds of developing sepsis (OR 1.38 [1.27–1.50]; OR 1.46 [1.25–1.70]), acute kidney injury (OR 1.13 [1.00–1.27]; OR 1.59 [1.32–1.91]), and ARDS (OR 1.38 [1.22–1.55]; OR 1.80 [1.49–2.17]). When comparing to stage 0, only stage 2 was associated with higher odds of hemodynamic (OR 1.46 [1.20–1.77]) and neurological (OR 1.58 [1.08–2.27]) complications.

**S1 Table** provides results of the sensitivity analysis that included only patients with lab-confirmed COVID-19. These results were largely consistent with the primary analysis.

**Table 4. Multivariable analysis for the association between LIC score and different outcomes (death, ICU admission, and IMV) among patients in the ISARIC clinical characterisation database (n = 17531).**

A. Odds ratio for the association between death and LIC score, adjusted for comorbidities, symptoms, and demographics.

| Term | Odds Ratio | 95% CI | P-value |
|---|---|---|---|
| LIC 0 | Ref | ref | ref |
| LIC 1 | 1.53 | (1.37–1.71) | <0.01 |
| LIC 2 | 2.50 | (2.1–2.96) | <0.01 |

B. Odds ratio for the association between ICU admission and LIC score, adjusted for comorbidities, symptoms, and demographics.

| Term | Odds Ratio | 95% CI | P-value |
|---|---|---|---|
| LIC 0 | Ref | ref | ref |
| LIC 1 | 1.63 | (1.48–1.79) | <0.01 |
| LIC 2 | 1.90 | (1.62–2.23) | <0.01 |

C. Odds ratio for the association between IMV treatment and LIC score, adjusted for comorbidities, symptoms, and demographics.

| Term | Odds Ratio | 95% CI | P-value |
|---|---|---|---|
| LIC 0 | Ref | ref | ref |
| LIC 1 | 1.43 | (1.27–1.70) | <0.01 |
| LIC 2 | 1.95 | (1.55–2.45) | <0.01 |

*The model was adjusted for: hematologic disease, chronic kidney disease, chronic neurological disorder, chronic pulmonary disease, dementia, diabetes, hypertension, liver disease, malignant neoplasm, obesity, smoking, age group, sex, cough, headache, shortness of breath, vomiting/nausea, and ICU admission.

*The model was adjusted for: AIDS/HIV, cardiac disease, pulmonary disease, asthma, chronic kidney disease, chronic neurological disorder, dementia, diabetes, hypertension, liver disease, obesity, malignant neoplasm, rheumatologic disorder, smoking, age group, sex, history of fever, shortness of breath.

*The model was adjusted for: AIDS/HIV, chronic cardiac disease, chronic hematologic disease, chronic kidney disease, chronic neurological disorder, chronic pulmonary disease, diabetes, hypertension, liver disease, malignant neoplasm, obesity, rheumatologic disorder, smoking, age group, sex, shortness of breath, vomiting nausea, ICU admission

**Table 5. Multivariable analysis for the association between in-hospital complications and liver injury score, while adjusted for demographics, comorbidities, treatments, and symptoms.** Reference category is stage 0 (n = 17531).

| Complication | Stage 0 (n = 8002) | Stage 1 (n = 8099) | Stage 2 (n = 1430) | OR (95% CI) for Stage 1 | OR (95% CI) for Stage 2 |
|---|---|---|---|---|---|
| Sepsis | | | | | |
| No | 4351 (54.4) | 3108 (38.4) | 495 (34.6) | 1.38 (1.27–1.5) | 1.46 (1.25–1.7) |
| Yes | 3274 (40.9) | 4550 (56.2) | 843 (59) | | |
| Unknown | 377 (4.7) | 441 (5.4) | 92 (6.4) | | |
| ARDS | | | | | |
| No | 6445 (80.5) | 5531 (68.3) | 896 (62.7) | 1.38 (1.22–1.55) | 1.8 (1.49–2.17) |
| Yes | 1003 (12.5) | 1864 (23) | 395 (27.6) | | |
| Neurological | | | | | |
| No | 7423 (92.8) | 7405 (91.4) | 1281 (89.6) | 0.95 (0.74–1.24) | 1.58 (1.08–2.27) |
| Yes | 144 (1.8) | 171 (2.1) | 48 (3.4) | | |
| Unknown | 435 (5.4) | 523 (6.5) | 101 (7.1) | | |
| Pulmonary | | | | | |
| No | 6324 (79) | 6120 (75.6) | 1069 (74.8) | 0.94 (0.79–1.11) | 1.13 (0.87–1.47) |
| Yes | 302 (3.8) | 399 (4.9) | 90 (6.3) | | |
| Unknown | 1376 (17.2) | 1580 (19.5) | 271 (19) | | |
| Hematological | | | | | |
| No | 6563 (82) | 6362 (78.6) | 1059 (74.1) | 0.85 (0.76–0.96) | 1.06 (0.88–1.27) |
| Yes | 995 (12.4) | 1193 (14.7) | 270 (18.9) | | |
| Unknown | 444 (5.5) | 544 (6.7) | 101 (7.1) | | |
| Hemodynamic | | | | | |
| No | 6788 (84.8) | 6427 (79.4) | 1059 (74.1) | 1.11 (0.98–1.26) | 1.46 (1.2–1.77) |
| Yes | 779 (9.7) | 1166 (14.4) | 274 (19.2) | | |
| Unknown | 435 (5.4) | 506 (6.2) | 97 (6.8) | | |
| Acute kidney injury | | | | | |
| No | 6630 (82.9) | 6211 (76.7) | 999 (69.9) | 1.13 (1.00–1.27) | 1.59 (1.32–1.91) |
| Yes | 893 (11.2) | 1271 (15.7) | 311 (21.7) | | |
| Unknown | 479 (6) | 617 (7.6) | 120 (8.4) | | |

Analyses adjusted for comorbidities, treatments, symptoms, and demographics.

## Discussion

Our analysis demonstrates that abnormalities in liver enzymes are common at admission in patients hospitalized with COVID-19. In our study, increasing severity of liver injury, as defined by abnormalities of transaminases or bilirubin, was associated with higher odds of mortality, ICU admission, and mechanical ventilation. Stage 1 and 2 were also associated with a higher odds of developing complications such as sepsis, AKI, and ARDS.

Previous smaller studies of patients with COVID-19 have shown similar results. Early data (n = 482) from China [20] found that nearly 30% of patients demonstrated liver enzyme abnormalities at baseline, with a higher unadjusted risk of mortality. In another small cohort (n = 147) from Germany [9], over 50% of patients had liver injury at baseline which was independently associated with mortality. In a larger cohort (n = 5771) of hospitalized patients from Hubei province [21], increasing values of AST, ALT, alkaline phosphatase, and bilirubin were associated with mortality, with AST the most deranged liver enzyme at admission for patients with severe COVID-19, and remaining high throughout hospitalization.

Results of our analysis are largely consistent with these prior studies and strengthen the evidence base with a much larger dataset. Multiple mechanisms may explain the frequency and

severity of liver involvement in COVID-19. The ubiquitous distribution of the viral entry receptor, ACE2, in human tissues might imply a role for direct cytopathic effects (22). Both microvesicular and macrovesicular steatosis has been demonstrated in autopsies of patients, with SARS-CoV-2 as the only risk factor for liver injury [22]. An additional component of hypoxic hepatitis in patients with severe hypoxic respiratory failure may also contribute [23]. Additional mechanisms include hepatic vascular thrombosis, widespread systemic inflammation, and drug-related toxicity [24, 25].

Our study has several important strengths: the large size of the cohort, the availability of data from multiple countries and sites, improving generalizability of findings, a pre-specified analysis plan, and adjustment for known confounders. Limitations of the analysis include the possibility of residual confounding, the inability to evaluate trends in liver enzymes during the course of hospital stay, and the impact of antivirals. In addition, bilirubin and transaminases do not assess similar aspects of liver injury, raising the possibility that our definition of liver injury was mis-specified. Selection bias is possible, since were able to include only a fraction of the patients represented in the ISARIC database due to missing data on liver enzyme tests and outcomes. Liver enzymes are more likely to be measured in patients with signs or history of liver disease, and in patients presenting with more severe illness at presentation, indicating that the frequency of liver failure reported here is an overestimate of the hospitalised population. Also, we were unable to evaluate the association between remdesivir and other antivirals on the development of liver injury due to extensive missing data.

## Conclusion

Liver enzyme abnormalities are common among COVID-19 patients and associated with worse outcomes. Multiple mechanisms may explain the extent and severity of liver injury in COVID-19. Future research should focus on understanding these mechanisms, the impact of changes over time, and whether antivirals improve or worsen liver injury.

## Supporting information

**S1 Table. Sensitivity analysis excluding non-PCR-confirmed SARS-CoV-2 patients.** Multivariable analysis for the association between LIC score and different outcomes (death, ICU admission, and IMV) among patients in the ISARIC clinical characterisation database (n = 17531). *The model was adjusted for: hematologic disease, chronic kidney disease, chronic neurological disorder, chronic pulmonary disease, dementia, diabetes, hypertension, liver disease, malignant neoplasm, obesity, smoking, age group, sex, cough, headache, shortness of breath, vomiting/nausea, and ICU admission.
(DOCX)

**S2 Table. ALT, AST, and Bilirubin relationship among patients in the ISARIC clinical characterisation database.** Normal upper limits (ULN) were taken as 1 mg/dL, 40 U/L, and 40 U/L for bilirubin, ALT, and AST, respectively.
(DOCX)

**S3 Table. STROBE checklist: This is the checklist for reporting observational studies.**
(DOCX)

**S1 Fig. Hospital Length of stay for the different stages of liver injury stratified by age and outcome.**
(TIF)

**S2 Fig. ICU length of stay for the different stages of liver injury stratified by age and out-come.**
(TIF)

**S1 File. ISARIC collaborators: This is the full list of collaborators along with their affilia-tions.**
(XLSX)

**S2 File. Statistical analysis plan: This is the prespecified and original statistical analysis plan for our submission.**
(DOCX)

## Acknowledgments

**ISARIC Clinical Characterisation Group**

Sheryl Ann Abdukahil, Nurul Najmee Abdulkadir, Ryuzo Abe, Laurent Abel, Amal Abrous, Lara Absil, Andrew Acker, Elisabeth Adam, Diana Adrião, Saleh Al Ageel, Kate Ainscough, Ali Ait Hssain, Younes Ait Tamlihat, Takako Akimoto, Ernita Akmal, Eman Al Qasim, Angela Alberti, Tala Al-dabbous, Senthilkumar Alegesan, Marta Alessi, Beatrice Alex, Kévin Alexandre, Abdulrahman Al-Fares, Huda Alfoudri, Imran Ali, Kazali Enagnon Alidjnou, Jeffrey Aliudin, Clotilde Allavena, Nathalie Allou, João Melo Alves, Rita Alves, Joana Alves Cabrita, Maria Amaral, Nur Amira, Phoebe Ampaw, Claire Andrejak, Andrea Angheben, François Angoulvant, Séverine Ansart, Sivanesen Anthonidass, Carlos Alexandre Antunes de Brito, Ardiyan Apriyana, Yaseen Arabi, Irene Aragao, Francisco Arancibia, Carolline Araujo, Antonio Arcadipane, Patrick Archambault, Lukas Arenz, Jean-Benoît Arlet, Christel Arnold-Day, Lovkesh Arora, Rakesh Arora, Elise Artaud-Macari, Diptesh Aryal, Muhammad Ashraf, Namra Asif, Jean Baptiste Assie, Amirul Asyraf, Minahel Atif, Anika Atique, Johann Auchabie, Hugues Aumaitre, Adrien Auvet, Laurène Azemar, Cecile Azoulay, Benjamin Bach, Delphine Bachelet, Claudine Badr, Nadia Baig, J. Kenneth Baillie, Erica Bak, Nazreen Abu Bakar, Andriy Bal, Mohanaprasanth Balakrishnan, Valeria Balan, Firouzé Bani-Sadr, Renata Barbalho, Nicholas Yuri Barbosa, Wendy S. Barclay, Saef Umar Barnett, Michaela Barnikel, Audrey Barrelet, Cleide Barrigoto, Marie Bartoli, Joaquín Baruch, Romain Basmaci, Muhammad Fadhli Hassin Basri, Jules Bauer, Diego Fernando Bautista Rincon, Abigail Beane, Alexandra Bedossa, Ker Hong Bee, Husna Begum, Sylvie Behilill, Albertus Beishuizen, Aleksandr Beljantsev, David Bellemare, Anna Beltrame, Beatriz Amorim Beltrão, Marine Beluze, Nicolas Benech, Lionel Eric Benjiman, Dehbia Benkerrou, Suzanne Bennett, Luís Bento, Jan-Erik Berdal, Delphine Bergeaud, Hazel Bergin, Giulia Bertoli, Simon Bessis, Sybille Bevilcaqua, Karine Bezulier, Amar Bhatt, Krishna Bhavsar, Claudia Bianco, Farah Nadiah Bidin, Moirangthem Bikram Singh, Felwa Bin Humaid, Mohd Nazlin Bin Kamarudin, François Bissuel, Patrick Biston, Laurent Bitker, Catherine Blier, Mathieu Blot, Lucille Blumberg, Laetitia Bodenes, Debby Bogaert, Anne-Hélène Boivin, Isabela Bolaños, Pierre-Adrien Bolze, François Bompart, Diogo Borges, Raphaël Borie, Elisabeth Botelho-Nevers, Lila Bouadma, Olivier Bouchaud, Sabelline Bouchez, Dounia Bouhmani, Damien Bouhour, Kévin Bouiller, Laurence Bouillet, Camile Bouisse, Anne-Sophie Boureau, John Bourke, Maude Bouscambert, Aurore Bousquet, Jason Bouziotis, Bianca Boxma, Marielle Boyer-Besseyre, Maria Boylan, Fernando Augusto Bozza, Axelle Braconnier, Cynthia Braga, Filipa Brás Monteiro, Luca Brazzi, Dorothy Breen, Kathy Brickell, Shaunagh Browne, Marjolein Brusse-Keizer, Petra Bryda, Nina Buchtele, Polina Bugaeva, Marielle Buisson, Erlina Burhan, Aidan Burrell, Ingrid G. Bustos, Denis Butnaru, André Cabie, Eder Caceres, Cyril Cadoz, Rui Caetano Garcês, Jose Andres Calvache, João Camões,

Valentine Campana, Paul Campbell, Cecilia Canepa, Pauline Caraux-Paz, Chiara Simona Cardellino, Filipa Cardoso, Filipe Cardoso, Nelson Cardoso, Sofia Cardoso, Nicolas Carlier, Thierry Carmoi, Gayle Carney, Inês Carqueja, Marie-Christine Carret, François Martin Carrier, Gail Carson, Maire-Laure Casanova, Mariana Cascão, Siobhan Casey, José Casimiro, Bailey Cassandra, Silvia Castañeda, Nidyanara Castanheira, Guylaine Castor-Alexandre, Ivo Castro, François-Xavier Catherine, Paolo Cattaneo, Roberta Cavalin, Alexandros Cavayas, Minerva Cervantes-Gonzalez, Anissa Chair, Catherine Chakveatze, Adrienne Chan, Meera Chand, Christelle Chantalat Auger, Jean-Marc Chapplain, Charlotte Charpentier, Julie Chas, Anjellica Chen, Yih-Sharng Chen, Léo Chenard, Matthew Pellan Cheng, Antoine Cheret, Thibault Chiarabini, Julian Chica, Suresh Kumar Chidambaram, Leong Chin Tho, Catherine Chirouze, Davide Chiumello, Bernard Cholley, Marie-Charlotte Chopin, Ting Soo Chow, Hiu Jian Chua, Jonathan Chua, Jose Pedro Cidade, Barbara Wanjiru Citarella, Anna Ciullo, Emma Clarke, Rolando Claure-Del Granado, Sara Clohisey, Cassidy Codan, Alexandra Coelho, Megan Coles, Gwenhaël Colin, Michael Collins, Sebastiano Maria Colombo, Pamela Combs, Jennifer Connolly, Marie Connor, Anne Conrad, Graham S. Cooke, Mary Copland, Hugues Cordel, Amanda Corley, Sabine Cornelis, Alexander Daniel Cornet, Arianne Joy Corpuz, Andrea Cortegiani, Grégory Corvaisier, Emma Costigan, Camille Couffignal, Sandrine Couffin-Cadiergues, Roxane Courtois, Stéphanie Cousse, Rachel Cregan, Sabine Croonen, Gloria Crowl, Jonathan Crump, Claudina Cruz, Marc Csete, Matthew Cummings, Elodie Curlier, Colleen Curran, Paula Custodio, Ana da Silva Filipe, Charlene Da Silveira, Al-Awwab Dabaliz, Andrew Dagens, Darren Dahly, Heidi Dalton, Jo Dalton, Seamus Daly, Juliana Damas, Nick Daneman, Corinne Daniel, Emmanuelle A Dankwa, Jorge Dantas, Etienne De Montmollin, Rafael Freitas de Oliveira França, Ana Isabel de Pinho Oliveira, Thushan de Silva, Peter de Vries, David Dean, Alexa Debard, Marie-Pierre Debray, Nathalie DeCastro, William Dechert, Lauren Deconninck, Romain Decours, Eve Defous, Isabelle Delacroix, Eric Delaveuve, Karen Delavigne, Nathalie M. Delfos, Ionna Deligiannis, Andrea Dell'Amore, Christelle Delmas, Pierre Delobel, Corine Delsing, Elisa Demonchy, Emmanuelle Denis, Dominique Deplanque, Pieter Depuydt, Mehul Desai, Diane Descamps, Mathilde Desvallées, Santi Dewayanti, Pathik Dhanger, Alpha Diallo, Sylvain Diamantis, Fernanda Dias Da Silva, Priscila Diaz, Rodrigo Diaz, Kévin Didier, Jean-Luc Diehl, Wim Dieperink, Jérôme Dimet, Vincent Dinot, Fara Diop, Alphonsine Diouf, Yael Dishon, Félix Djossou, Annemarie B. Docherty, Christl A. Donnelly, Chloe Donohue, Sean Donohue, Yoann Donohue, Peter Doran, Céline Dorival, Eric D'Ortenzio, James Joshua Douglas, Nathalie Dournon, Triona Downer, Joanne Downey, Mark Downing, Tom Drake, Murray Dryden, Claudio Duarte Fonseca, Vincent Dubee, François Dubos, Alexandre Ducancelle, Toni Duculan, Susanne Dudman, Abhijit Duggal, Paul Dunand, Jake Dunning, Mathilde Duplaix, Lucian Durham III, Bertrand Dussol, Juliette Duthoit, Xavier Duval, Anne Margarita Dyrhol-Riise, Sim Choon Ean, Marco Echeverria-Villalobos, Carla Eira, Mohammed El Sanharawi, Subbarao Elapavaluru, Brigitte Elharrar, Jacobien Ellerbroek, Philippine Eloy, Tarek Elshazly, Isabelle Enderle, Chan Chee Eng, Ilka Engelmann, Vincent Enouf, Olivier Epaulard, Martina Escher, Mariano Esperatti, Hélène Esperou, Catarina Espírito Santo, Marina Esposito-Farese, João Estevão, Manuel Etienne, Nadia Ettalhaoui, Anna Greti Everding, Mirjam Evers, Isabelle Fabre, Marc Fabre, Amna Faheem, Cameron J. Fairfield, Pedro Faria, Ahmed Farooq, Hanan Fateena, Arie Zainul Fatoni, Karine Faure, Raphaël Favory, Mohamed Fayed, Laura Feeney, Jorge Fernandes, Marília Andreia Fernandes, Susana Fernandes, François-Xavier Ferrand, Eglantine Ferrand Devouge, Joana Ferrão, Mário Ferraz, Benigno Ferreira, Bernardo Ferreira, Isabel Ferreira, Sílvia Ferreira, Nicolas Ferriere, Céline Ficko, Claudia Figueiredo-Mello, Juan Fiorda, Thomas Flament, Clara Flateau, Tom Fletcher, Aline-Marie Florence, Deirdre Flynn, Jean Foley, Victor Fomin, Tatiana Fonseca, Simon Forsyth, Giuseppe Foti, Erwan Fourn, Robert A. Fowler,

Marianne Fraher, Diego Franch-Llasat, Christophe Fraser, John F Fraser, Marcela Vieira Freire, Ana Freitas Ribeiro, Craig French, Caren Friedrich, Ricardo Fritz, Stéphanie Fry, Nora Fuentes, Masahiro Fukuda, Argin G, Valérie Gaborieau, Rostane Gaci, Jean-Charles Gagnard, Amandine Gagneux-Brunon, Sérgio Gaião, Linda Gail Skeie, Phil Gallagher, Carrol Gamble, Yasmin Gani, Arthur Garan, Rebekha Garcia, Esteban Garcia-Gallo, Denis Garot, Valérie Garrait, Nathalie Gault, Aisling Gavin, Anatoliy Gavrylov, Alexandre Gaymard, Johannes Gebauer, Eva Geraud, Louis Gerbaud Morlaes, Nuno Germano, Jade Ghosn, Marco Giani, Carlo Giaquinto, Tristan Gigante, Morgane Gilg, Guillermo Giordano, Michelle Girvan, Valérie Gissot, Daniel Glikman, Petr Glybochko, Eric Gnall, François Goehringer, Siri Goepel, Jean-Christophe Goffard, Jin Yi Goh, Jonathan Golob, Joan Gómez-Junyent, Marie Gominet, Alicia Gonzalez, Patricia Gordon, Isabelle Gorenne, Laure Goubert, Cécile Goujard, Tiphaine Goulenok, Margarite Grable, Edward Wilson Grandin, Pascal Granier, Giacomo Grasselli, Christopher A. Green, Courtney Greene, William Greenhalf, Segolène Greffe, Matthew Griffee, Fiona Griffiths, Albert Groenendijk, Anja Grosse Lordemann, Heidi Gruner, Yusing Gu, Jérémie Guedj, Martin Guego, Dewi Guellec, Daniela Guerreiro, Romain Guery, Anne Guillaumot, Laurent Guilleminault, Maisa Guimarães de Castro, Thomas Guimard, Marieke Haalboom, Daniel Haber, Ali Hachemi, Nadir Hadri, Adam Hall, Matthew Hall, Sophie Halpin, Ansley Hamer, Rebecca Hamidfar, Terese Hammond, Lim Yuen Han, Rashan Haniffa, Kok Wei Hao, Hayley Hardwick, Ewen M. Harrison, Janet Harrison, Alan Hartman, Ailbhe Hayes, Leanne Hays, Jan Heerman, Lars Heggelund, Ross Hendry, Martina Hennessy, Aquiles Henriquez, Maxime Hentzien, Diana Hernandez, Andrew Hershey, Liv Hesstvedt, Dawn Higgins, Eibhlin Higgins, Rupert Higgins, Samuel Hinton, Hiroaki Hiraiwa, Hikombo Hitoto, Antonia Ho, Yi Bin Ho, Alexandre Hoctin, Isabelle Hoffmann, Wei Han Hoh, Oscar Hoiting, Jan Cato Holter, Peter Horby, Juan Pablo Horcajada, Koji Hoshino, Kota Hoshino, Ikram Houas, Catherine L. Hough, Stuart Houltham, Jimmy Ming-Yang Hsu, Jean-Sébastien Hulot, Stella Huo, Abby Hurd, Iqbal Hussain, Samreen Ijaz, Arfan Ikram, Hajnal-Gabriela Illes, Patrick Imbert, Rana Imran Sikander, Hugo Inácio, Yun Sii Ing, Elias Iosifidis, Mariachiara Ippolito, Vera Irawany, Sarah Isgett, Tiago Isidoro, Nadiah Ismail, Margaux Isnard, Daniel Ivulich, Danielle Jaafar, Salma Jaafoura, Julien Jabot, Clare Jackson, Nina Jamieson, Pierre Jaquet, Waasila Jassat, Coline Jaud-Fischer, Stéphane Jaureguiberry, Florence Jego, Anilawati Mat Jelani, Synne Jenum, Ruth Jimbo-Sotomayor, Ong Yiaw Joe, Ruth N. Jorge García, Cédric Joseph, Mark Joseph, Mercé Jourdain, Anna Jung, Dafsah Juzar, Ouifiya Kafif, Florentia Kaguelidou, Neerusha Kaisbain, Thavamany Kaleesvran, Sabina Kali, Muhammad Aisar Ayadi Kamaluddin, Zul Amali Che Kamaruddin, Nadiah Kamarudin, Chris Kandel, Kong Yeow Kang, Pratap Karpayah, Christiana Kartsonaki, Daisuke Kasugai, Anant Kataria, Kevin Katz, Christy Kay, Lamees Kayyali, Hannah Keane, Seán Keating, Andrea Kelly, Claire Kelly, Niamh Kelly, Sadie Kelly, Maeve Kelsey, Kalynn Kennon, Maeve Kernan, Younes Kerroumi, Sharma Keshav, Imrana Khalid, Antoine Khalil, Coralie Khan, Irfan Khan, Sushil Khanal, Michelle E Kho, Denisa Khoo, Ryan Khoo, Saye Khoo, Khor How Kiat, Yuri Kida, Peter Kiiza, Beathe Kiland Granerud, Jae Burm Kim, Antoine Kimmoun, Alexander King, Nobuya Kitamura, Paul Klenerman, Rob Klont, Gry Kloumann Bekken, Stephen R Knight, Robin Kobbe, Caroline Kosgei, Arsène Kpangon, Karolina Krawczyk, Sudhir Krishnan, Vinothini Krishnan, Oksana Kruglova, Deepali Kumar, Ganesh Kumar, Bharath Kumar Tirupakuzhi Vijayaraghavan, Pavan Kumar Vecham, Ethan Kurtzman, Demetrios Kutsogiannis, Galyna Kutsyna, Marie Lachatre, Marie Lacoste, John G. Laffey, Nadhem Lafhej, Marie Lagrange, Fabrice Laine, Olivier Lairez, Marc Lambert, Marie Langelot-Richard, Vincent Langlois, Eka Yudha Lantang, Marina Lanza, Cédric Laouénan, Samira Laribi, Delphine Lariviere, Stéphane Lasry, Naveed Latif, Odile Launay, Didier Laureillard, Yoan Lavie-Badie, Andrew Law, Teresa Lawrence, Minh Le, Clément Le Bihan, Cyril Le Bris, Georges Le Falher, Lucie Le Fevre, Quentin Le Hingrat, Marion Le

Maréchal, Soizic Le Mestre, Gwenaël Le Moal, Vincent Le Moing, Hervé Le Nagard, Paul Le Turnier, Biing Horng Lee, Heng Gee Lee, James Lee, Jennifer Lee, Su Hwan Lee, Todd C. Lee, Yi Lin Lee, Gary Leeming, Bénédicte Lefebvre, Laurent Lefebvre, Benjamin Lefèvre, Sylvie LeGac, Jean-Daniel Lelievre, François Lellouche, Adrien Lemaignen, Véronique Lemee, Anthony Lemeur, Gretchen Lemmink, Ha Sha Lene, Jenny Lennon, Marc Leone, Michela Leone, Quentin Lepiller, François-Xavier Lescure, Olivier Lesens, Mathieu Lesouhaitier, Amy Lester-Grant, Sophie Letrou, Bruno Levy, Yves Levy, Claire Levy-Marchal, Katarzyna Lewandowska, Erwan L'Her, Gianluigi Li Bassi, Geoffrey Liegeon, Kah Chuan Lim, Wei Shen Lim, Chantre Lima, Bruno Lina, Lim Lina, Andreas Lind, Guillaume Lingas, Sylvie Lion-Daolio, Samantha Lissauer, Marine Livrozet, Patricia Lizotte, Navy Lolong, Leong Chee Loon, Diogo Lopes, Anthony L. Loschner, Paul Loubet, Bouchra Loufti, Guillame Louis, Silvia Lourenco, Lee Lee Low, Marije Lowik, Jia Shyi Loy, Jean Christophe Lucet, Carlos M. Luna, Liem Luong, Nestor Luque, Dominique Luton, Nilar Lwin, Ruth Lyons, Olavi Maasikas, Oryane Mabiala, Moïse Machado, Sara Machado, Gabriel Macheda, Hashmi Madiha, Rafael Mahieu, Sophie Mahy, Ana Raquel Maia, Lars S. Maier, Mylène Maillet, Thomas Maitre, Maximilian Malfertheiner, Nadia Malik, Paddy Mallon, Denis Malvy, Victoria Manda, Laurent Mandelbrot, Frank Manetta, Julie Mankikian, Edmund Manning, Aldric Manuel, Ceila Maria Sant'Ana Malaque, Flávio Marino, Samuel Markowicz, Charbel Maroun Eid, Laura Marsh, John Marshall, Celina Turchi Martelli, Dori-Ann Martin, Emily Martin, Guillaume Martin-Blondel, Ignacio Martin-Loeches, Martin Martinot, Alejandro Martín-Quiros, Ana Martins, Nuno Martins, Caroline Martins Rego, Gennaro Martucci, Olga Martynenko, Eva Miranda Marwali, Marsilla Marzukie, David Maslove, Basri Mat Nor, Moshe Matan, Daniel Mathieu, Mathieu Mattei, Romans Matulevics, Laurence Maulin, Michael Maxwell, Thierry Mazzoni, Lisa Mc Sweeney, Colin McArthur, Aine McCarthy, Anne McCarthy, Colin McCloskey, Sherry McDermott, Sarah E. McDonald, Allison McGeer, Johnny McKeown, Kenneth A. McLean, Paul McNally, Bairbre McNicholas, Elaine McPartlan, Edel Meaney, Cécile Mear-Passard, Maggie Mechlin, Omar Mehkri, Luis Melo, Joao Joao Mendes, France Mentré, Alexander J. Mentzer, Emmanuelle Mercier, Noémie Mercier, Antoine Merckx, Mayka Mergeay-Fabre, Blake Mergler, Laura Merson, António Mesquita, Osama Metwally, Agnès Meybeck, Dan Meyer, Alison M. Meynert, Vanina Meysonnier, Amina Meziane, Mehdi Mezidi, Céline Michelanglei, Isabelle Michelet, Efstathia Mihelis, Vladislav Mihnovit, Hugo Miranda-Maldonado, Nor Arisah Misnan, Nik Nur Eliza Mohamed, Tahira Jamal Mohamed, Asma Moin, Elena Molinos, Brenda Molloy, Sinead Monahan, Mary Mone, Agostinho Monteiro, Claudia Montes, Giorgia Montrucchio, Sarah Moore, Shona C. Moore, Lina Morales Cely, Lucia Moro, Ben Morton, Ana Motos, Hugo Mouquet, Clara Mouton Perrot, Julien Moyet, Caroline Mudara, Ng Yong Muh, Dzawani Muhamad, Jimmy Mullaert, Fredrik Müller, Karl Erik Müller, Daniel Munblit, Aisling Murphy, Lorna Murphy, Patrick Murray, Marlène Murris, Srinivas Murthy, Himed Musaab, Gugapriyaa Muyandy, Alex Nagrebetsky, Mangala Narasimhan, Alasdair Nazerali-Maitland, Nadège Neant, Holger Neb, Nikita Nekliudov, Raul Neto, Emily Neumann, Anthony Nghi, Duc Nguyen, Niamh Ni Leathlobhair, Alistair Nichol, Prompak Nitayavardhana, Stephanie Nonas, Nurul Amani Mohd Noordin, Marion Noret, Nurul Faten Izzati Norharizam, Lisa Norman, Alessandra Notari, Mahdad Noursadeghi, Adam Nowinski, Saad Nseir, Jose I Nunez, Elsa Nyamankolly, Fionnuala O Brien, Annmarie O Callaghan, Annmarie O'Callaghan, Giovanna Occhipinti, Derbrenn OConnor, Max O'Donnell, Tawnya Ogston, João Oliveira, Larissa Oliveira, Piero L. Olliaro, Conar O'Neil, David S.Y. Ong, Jee Yan Ong, Wilna Oosthuyzen, Peter Openshaw, Claudia Milena Orozco-Chamorro, Jamel Ortoleva, Javier Osatnik, Linda O'Shea, Miriam O'Sullivan, Siti Zubaidah Othman, Nadia Ouamara, Rachida Ouissa, Eric Oziol, Maïder Pagadoy, Justine Pages, Amanda Palacios, Massimo Palmarini, Giovanna Panarello, Prasan Kumar Panda, Lai Hui Pang, Mauro Panigada,

Nathalie Pansu, Aurélie Papadopoulos, Rachael Parke, Jérémie Pasquier, Bruno Pastene, Mohan Dass Pathmanathan, Luís Patrão, Patricia Patricio, Juliette Patrier, Lisa Patterson, Christelle Paul, Mical Paul, Jorge Paulos, William A. Paxton, Jean-François Payen, Kalaiarasu Peariasamy, Florent Peelman, Nathan Peiffer-Smadja, Vincent Peigne, Mare Pejkovska, Ithan D. Peltan, Rui Pereira, Daniel Perez, Luis Periel, Thomas Perpoint, Antonio Pesenti, Vincent Pestre, Michele Petrovic, Ventzislava Petrov-Sanchez, Frank Olav Pettersen, Gilles Peytavin, Scott Pharand, Michael Piagnerelli, Walter Picard, Olivier Picone, Maria de Piero, Carola Pierobon, Djura Piersma, Carlos Pimentel, Valentine Piquard, Catarina Pires, Isabelle Pironneau, Lionel Piroth, Riinu Pius, Laurent Plantier, Hon Shen Png, Julien Poissy, Ryadh Pokeerbux, Sergio Poli, Georgios Pollakis, Diane Ponscarme, Diego Bastos Porto, Andra-Maris Post, Douwe F. Postma, Pedro Povoa, Diana Póvoas, Jeff Powis, Sébastien Preau, Christian Prebensen, Jean-Charles Preiser, Anton Prinssen, Mark G. Pritchard, Lucia Proença, Oriane Puéchal, Bambang Pujo Semedi, Gregory Purcell, Luisa Quesada, Else Quist-Paulsen, Mohammed Quraishi, Fadi-Fadi Qutishat, Christian Rabaud, Aldo Rafael, Marie Rafiq, Ahmad Kashfi Haji Ab Rahman, Rozanah Abd Rahman, Fernando Rainieri, Giri Shan Rajahram, Nagarajan Ramakrishnan, José Ramalho, Ahmad Afiq Ramli, Blandine Rammaert, Grazielle Viana Ramos, Asim Rana, Ritika Ranjan, Christophe Rapp, Menaldi Rasmin, Indrek Rätsep, Cornelius Rau, Tharmini Ravi, Stanislas Rebaudet, Sarah Redl, Brenda Reeve, Liadain Reid, Dag Henrik Reikvam, Renato Reis, Jonathan Remppis, Martine Remy, Hongru Ren, Hanna Renk, Anne-Sophie Resseguier, Matthieu Revest, Oleksa Rewa, Luis Felipe Reyes, Tiago Reyes, Maria Ines Ribeiro, David Richardson, Denise Richardson, Laurent Richier, Siti Nurul Atikah Ahmad Ridzuan, Ana L Rios, Asgar Rishu, Patrick Rispal, Karine Risso, Maria Angelica Rivera Nuñez, Nicholas Rizer, André Roberto, Stephanie Roberts, David L. Robertson, Olivier Robineau, Ferran Roche-Campo, Paola Rodari, Simão Rodeia, Bernhard Roessler, Pierre-Marie Roger, Emmanuel Roilides, Amanda Rojek, Juliette Romaru, Roberto Roncon-Albuquerque Jr, Mélanie Roriz, Manuel Rosa-Calatrava, Michael Rose, Dorothea Rosenberger, Andrea Rossanese, Matteo Rossetti, Bénédicte Rossignol, Patrick Rossignol, Stella Rousset, Carine Roy, Benoît Roze, Clark D. Russell, Maeve Ryan, Steffi Ryckaert, Aleksander Rygh Holten, Isabela Saba, Musharaf Sadat, Valla Sahraei, Maximilien Saint-Gilles, Pranya Sakiyalak, Leonardo Salazar, Gabriele Sales, Stéphane Sallaberry, Charlotte Salmon Gandonniere, Hélène Salvator, Olivier Sanchez, Vanessa Sancho-Shimizu, Gyan Sandhu, Zulfiqar Sandhu, Pierre-François Sandrine, Oana Sandulescu, Marlene Santos, Shirley Sarfo-Mensah, Bruno Sarmento Banheiro, Iam Claire E. Sarmiento, Benjamine Sarton, Sree Satyapriya, Rumaisah Satyawati, Egle Saviciute, Parthena Savvidou, Yen Tsen Saw, Tjard Schermer, Arnaud Scherpereel, Marion Schneider, Michael Schwameis, Janet T. Scott, James Scott-Brown, Nicholas Sedillot, Mageswari Selvarajoo, Caroline Semaille, Malcolm G. Semple, Rasidah Bt Senian, Eric Senneville, Claudia Sepulveda, Filipa Sequeira, Tânia Sequeira, Ary Serpa Neto, Ellen Shadowitz, Syamin Asyraf Shahidan, Mohammad Shamsah, Shaikh Sharjeel, Pratima Sharma, Catherine A. Shaw, Victoria Shaw, John Robert Sheenan, Haixia Shi, Hiroaki Shimizu, Keiki Shimizu, Sally Shrapnel, Nassima Si Mohammed, Ng Yong Siang, Jeanne Sibiude, Louise Sigfrid, Piret Sillaots, Maria Joao Silva, Rogério Silva, Benedict Sim Lim Heng, Budha Charan Singh, Pompini Agustina Sitompul, Karisha Sivam, Sue Smith, Benjamin Smood, Michelle Smyth, Morgane Snacken, Tze Vee Soh, Joshua Solomon, Tom Solomon, Emily Somers, Agnès Sommet, Myung Jin Song, Tae Song, Jack Song Chia, Michael Sonntagbauer, Azlan Mat Soom, Albert Sotto, Edouard Soum, Ana Chora Sousa, Marta Sousa, Maria Sousa Uva, Vicente Souza-Dantas, Alexandra Sperry, Elisabetta Spinuzza, Shiranee Sriskandan, Sarah Stabler, Thomas Staudinger, Stephanie-Susanne Stecher, Trude Steinsvik, Ymkje Stienstra, Birgitte Stiksrud, Amy Stone, Adrian Streinu-Cercel, Anca Streinu-Cercel, Ami Stuart, David Stuart, Richa Su, Gabriel Suen, Jacky Y. Suen, Prasanth Sukumar, Asfia Sultana, Charlotte Summers,

Deepashankari Suppiah, Andrey Svistunov, Sarah Syahrin, Jaques Sztajnbok, Konstanty Szuldrzynski, Shirin Tabrizi, Fabio S. Taccone, Lysa Tagherset, Shahdattul Mawarni Taib, Sara Taleb, Jelmer Talsma, Maria Lawrensia Tampubolon, Kim Keat Tan, Yan Chyi Tan, Taku Tanaka, Coralie Tardivon, Pierre Tattevin, Hassan Tawfik, Richard S. Tedder, Tze Yuan Tee, João Teixeira, Marie-Capucine Tellier, Sze Kye Teoh, François Téoulé, Pleun Terpstra, Olivier Terrier, Nicolas Terzi, Hubert Tessier-Grenier, Alif Adlan Mohd Thabit, Zhang Duan Tham, Suvintheran Thangavelu, Elmi Theron, Vincent Thibault, Simon-Djamel Thiberville, Benoît Thill, Jananee Thirumanickam, Shaun Thompson, David Thomson, Emma C. Thomson, Surain Raaj Thanga Thurai, Ryan S. Thwaites, Vadim Tieroshyn, Peter S Timashev, Jean-François Timsit, Noémie Tissot, Fiona Toal, Jordan Zhien Yang Toh, Kristian Tonby, Sia Loong Tonnii, Marta Torre, Antoni Torres, Rosario Maria Torres Santos-Olmo, Hernando Torres-Zevallos, Tony Trapani, Théo Trioux, Cécile Tromeur, Tiffany Trouillon, Jeanne Truong, Christelle Tual, Sarah Tubiana, Helen Tuite, Jean-Marie Turmel, Lance C.W. Turtle, Anders Tveita, Makoto Uchiyama, Andrew Udy, Roman Ullrich, Alberto Uribe, Asad Usman, Timothy M. Uyeki, Luís Val-Flores, Amélie Valran, Stijn Van de Velde, Marcel van den Berge, Machteld Van der Feltz, Job van der Palen, Paul van der Valk, Nicky Van Der Vekens, Peter Van der Voort, Sylvie Van Der Werf, Laura van Gulik, Jarne Van Hattem, Carolien van Netten, Ilonka van Veen, Noémie Vanel, Henk Vanoverschelde, Pooja Varghese, Michael Varrone, Shoban Raj Vasudayan, Charline Vauchy, Shaminee Veeran, Aurélie Veislinger, Sebastian Vencken, Sara Ventura, Annelies Verbon, James Vickers, José Ernesto Vidal, César Vieira, Deepak Vijayan, Joy Ann Villanueva, Judit Villar, Pierre-Marc Villeneuve, Andrea Villoldo, Benoit Visseaux, Chiara Vitiello, Harald Vonkeman, Fanny Vuotto, Noor Hidayu Wahab, Suhaila Abdul Wahab, Nadirah Abdul Wahid, Marina Wainstein, Laura Walsh, Chih-Hsien Wang, Steve Webb, Jia Wei, Tan Pei Wen, Sanne Wesselius, Murray Wham, Bryan Whelan, Nicole White, Paul Henri Wicky, Aurélie Wiedemann, Surya Otto Wijaya, Suzette Willems, Virginie Williams, Evert-Jan Wils, Calvin Wong, Teck Fung Wong, Xin Ci Wong, Yew Sing Wong, Gan Ee Xian, Lim Saio Xian, Kuan Pei Xuan, Siti Rohani Binti Mohd Yakop, Masaki Yamazaki, Yazdan Yazdanpanah, Nicholas Yee Liang Hing, Cécile Yelnik, Chian Hui Yeoh, Stephanie Yerkovich, Toshiki Yokoyama, Hodane Yonis, Obada Yousif, Akram Zaaqoq, Marion Zabbe, Kai Zacharowski, Masliza Zahid, Maram Zahran, Nor Zaila Binti Zaidan, Maria Zambon, Miguel Zambrano, Alberto Zanella, Nurul Zaynah, Hiba Zayyad, David Zucman.

The investigators acknowledge the support of the COVID clinical management team, AIIMS, Rishikesh, India; the dedication and hard work of the Groote Schuur Hospital Covid ICU Team and supported by the Groote Schuur nursing and University of Cape Town registrar bodies coordinated by the Division of Critical Care at the University of Cape Town; the Liverpool School of Tropical Medicine and the University of Oxford; Imperial NIHR Biomedical Research Centre; the dedication and hard work of the Norwegian SARS-CoV-2 study team; endorsement of the Irish Critical Care- Clinical Trials Group, co-ordination in Ireland by the Irish Critical Care- Clinical Trials Network at University College Dublin; and preparedness work conducted by the Short Period Incidence Study of Severe Acute Respiratory Infection.

This work uses data provided by patients and collected by the NHS as part of their care and support #DataSavesLives. The data used for this research were obtained from ISARIC4C. We are extremely grateful to the 2648 frontline NHS clinical and research staff and volunteer medical students who collected these data in challenging circumstances; and the generosity of the patients and their families for their individual contributions in these difficult times. The COVID-19 Clinical Information Network (CO-CIN) data was collated by ISARIC4C Investigators. We also acknowledge the support of Jeremy J Farrar and Nahoko Shindo.

## Author Contributions

**Conceptualization:** Bharath Kumar Tirupakuzhi Vijayaraghavan, Saptarshi Bishnu, Joaquin Baruch, Neill K. J. Adhikari.

**Data curation:** Barbara Wanjiru Citarella, Laura Merson.

**Formal analysis:** Bharath Kumar Tirupakuzhi Vijayaraghavan, Saptarshi Bishnu, Joaquin Baruch, Aronrag Meeyai, Neill K. J. Adhikari.

**Investigation:** Bharath Kumar Tirupakuzhi Vijayaraghavan, Saptarshi Bishnu, Joaquin Baruch, Barbara Wanjiru Citarella, Christiana Kartsonaki, Zubair Mohamed, Shinichiro Ohshimo, Benjamin Lefèvre, Abdulrahman Al-Fares, Jose A. Calvache, Fabio Silvio Taccone, Piero Olliaro, Laura Merson, Neill K. J. Adhikari.

**Methodology:** Bharath Kumar Tirupakuzhi Vijayaraghavan, Saptarshi Bishnu, Joaquin Baruch, Aronrag Meeyai, Neill K. J. Adhikari.

**Project administration:** Barbara Wanjiru Citarella, Laura Merson.

**Supervision:** Neill K. J. Adhikari.

**Writing – original draft:** Bharath Kumar Tirupakuzhi Vijayaraghavan, Saptarshi Bishnu, Joaquin Baruch, Neill K. J. Adhikari.

**Writing – review & editing:** Bharath Kumar Tirupakuzhi Vijayaraghavan, Saptarshi Bishnu, Joaquin Baruch, Barbara Wanjiru Citarella, Christiana Kartsonaki, Aronrag Meeyai, Zubair Mohamed, Shinichiro Ohshimo, Benjamin Lefèvre, Abdulrahman Al-Fares, Jose A. Calvache, Fabio Silvio Taccone, Piero Olliaro, Laura Merson, Neill K. J. Adhikari.

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
