## [Decision Letter · Decision Letter 0]

23 Jun 2023

PONE-D-22-30411Liver injury in hospitalized patients with COVID-19: An International observational cohort studyPLOS ONE

Dear Dr. Tirupakuzhi Vijayaraghavan,

Thank you for submitting your manuscript to PLOS ONE. After careful consideration, we feel that it has merit but does not fully meet PLOS ONE’s publication criteria as it currently stands. Therefore, we invite you to submit a revised version of the manuscript that addresses the points raised during the review process.

We look forward to receiving your revised manuscript.

Kind regards,

Aleksandar R. Zivkovic

Academic Editor

PLOS ONE

Journal Requirements:

3. Thank you for stating the following in the Acknowledgments/ Funding Section of your manuscript:

“This work was made possible by the UK Foreign, Commonwealth and Development Office

and Wellcome [215091/Z/18/Z, 222410/Z/21/Z, 225288/Z/22/Z and 220757/Z/20/Z]; the Bill

& Melinda Gates Foundation [OPP1209135]; the philanthropic support of the donors to the

University of Oxford’s COVID-19 Research Response Fund (0009109); CIHR Coronavirus

Rapid Research Funding Opportunity OV2170359 and the coordination in Canada by

Sunnybrook Research Institute; endorsement of the Irish Critical Care- Clinical Trials Group,

co-ordination in Ireland by the Irish Critical Care- Clinical Trials Network at University

College Dublin and funding by the Health Research Board of Ireland [CTN-2014-12]; the

COVID clinical management team, AIIMS, Rishikesh, India; Cambridge NIHR Biomedical

Research Centre; the dedication and hard work of the Groote Schuur Hospital Covid ICU

Team and supported by the Groote Schuur nursing and University of Cape Town registrar

bodies coordinated by the Division of Critical Care at the University of Cape Town; the

Liverpool School of Tropical Medicine and the University of Oxford; the dedication and hard

work of the Norwegian SARS-CoV-2 study team and the Research Council of Norway grant

no 312780, and a philanthropic donation from Vivaldi Invest A/S owned by Jon Stephenson

von Tetzchner; Imperial NIHR Biomedical Research Centre; the Comprehensive Local

Research Networks (CLRNs) of which PJMO is an NIHR Senior Investigator

(NIHR201385); Innovative Medicines Initiative Joint Undertaking under Grant Agreement

No. 115523 COMBACTE, resources of which are composed of financial contribution from

the European Union’s Seventh Framework Programme (FP7/2007- 2013) and EFPIA

companies, in-kind contribution; Stiftungsfonds zur Förderung der Bekämpfung der

Tuberkulose und anderer Lungenkrankheiten of the City of Vienna, Project Number:

APCOV22BGM; Italian Ministry of Health “Fondi Ricerca corrente–L1P6” to IRCCS

Ospedale Sacro Cuore–Don Calabria; Australian Department of Health grant (3273191);

Gender Equity Strategic Fund at University of Queensland, Artificial Intelligence for

Pandemics (A14PAN) at University of Queensland, the Australian Research Council Centre

of Excellence for Engineered Quantum Systems (EQUS, CE170100009), the Prince Charles

Hospital Foundation, Australia; Brazil, National Council for Scientific and Technological

Development Scholarship number 303953/2018- 7; the Firland Foundation, Shoreline,

Washington, USA;  the French COVID cohort (NCT04262921) is sponsored by INSERM

and is funded by the REACTing (REsearch & ACtion emergING infectious diseases)

consortium and by a grant of the French Ministry of Health (PHRC n°20-0424); a grant from

foundation Bevordering Onderzoek Franciscus; the South Eastern Norway Health Authority

and the Research Council of Norway; Institute for Clinical Research (ICR), National

Institutes of Health (NIH) supported by the Ministry of Health Malaysia; preparedness work

conducted by the Short Period Incidence Study of Severe Acute Respiratory Infection.

This work uses data provided by patients and collected by the NHS as part of their care and

support #DataSavesLives. The data used for this research were obtained from ISARIC4C. We

are extremely grateful to the 2648 frontline NHS clinical and research staff and volunteer

medical students who collected these data in challenging circumstances; and the generosity of

the patients and their families for their individual contributions in these difficult times. The

COVID-19 Clinical Information Network (CO-CIN) data was collated by ISARIC4C

Investigators. Data and Material provision was supported by grants from: the National

Institute for Health Research (NIHR; award CO-CIN-01), the Medical Research Council

(MRC; grant MC_PC_19059), and by the NIHR Health Protection Research Unit (HPRU) in

Emerging and Zoonotic Infections at University of Liverpool in partnership with Public

Health England (PHE), (award 200907), NIHR HPRU in Respiratory Infections at Imperial

College London with PHE (award 200927), Liverpool Experimental Cancer Medicine Centre

(grant C18616/A25153), NIHR Biomedical Research Centre at Imperial College London

(award ISBRC-1215-20013), and NIHR Clinical Research Network providing infrastructure

support. We also acknowledge the support of Jeremy J Farrar and Nahoko Shindo.”

“Recipient: Funding received by the International Severe Acute Respiratory and emerging Infection Consortium (ISARIC).

This work was made possible by the UK Foreign, Commonwealth and Development Office and Wellcome [215091/Z/18/Z, 222410/Z/21/Z, 225288/Z/22/Z and 220757/Z/20/Z]; the Bill

& Melinda Gates Foundation [OPP1209135];

URL:

https://wellcome.org/

https://www.gov.uk/government/organisations/foreign-commonwealth-development-office

The funders had no role in study design, data collection and analysis, decision to publish, or preparation of the manuscript”

6. "One of the noted authors is a group or consortium [ISARIC Clinical Characterisation Group]. In addition to naming the author group, please list the individual authors and affiliations within this group in the acknowledgments section of your manuscript. Please also indicate clearly a lead author for this group along with a contact email address."

Reviewers' comments:

Reviewer's Responses to Questions

**Comments to the Author**

1. Is the manuscript technically sound, and do the data support the conclusions?

Reviewer #1: Partly

2. Has the statistical analysis been performed appropriately and rigorously? 

Reviewer #1: I Don't Know

3. Have the authors made all data underlying the findings in their manuscript fully available?

Reviewer #1: Yes

4. Is the manuscript presented in an intelligible fashion and written in standard English?

Reviewer #1: No

5. Review Comments to the Author

Reviewer #1: Well done! Very interesting findings that can contribute to more effective way in managing COVID-19 patients. There are some formatting and standard English errors that need to be reviewed. Some of the information can be misleading and confusing. The conclusion needs to be more impactful.

6. PLOS authors have the option to publish the peer review history of their article (what does this mean?). If published, this will include your full peer review and any attached files.

Reviewer #1: No

---

## [Author Response · Author response to Decision Letter 0]

27 Jul 2023

We thank the Editor and the Reviewers for the feedback. The comments and suggestions have helped us improve our manuscript and we are grateful for this. We provide a point-by-point response to each of the comments. Editor and reviewer comments are italicized, and our responses are listed below each comment.

Response: Thank you. We have checked and revised to meet these requirements.

Response: Our manuscript is a secondary analysis of data available in the International Severe Acute Respiratory and emerging Infection Consortium(ISARIC) COVID-19 dataset. As such, direct patient-level consent is not applicable for this analysis, but was obtained where applicable and required for primary data collection and reuse. At the onset of the pandemic, ISARIC in collaboration with the World Health Organization developed a clinical characterization protocol and a standardized case report form (CRF) for harmonized collection of data across hospitals and countries. The CRF was made available as an open-access resource and a central REDCAP database was hosted by ISARIC. Sites across the world had the option of contributing data to this central database after obtaining appropriate local ethics approvals and patient-level consent (if deemed necessary as per site and country requirements). 

The ISARIC-WHO Clinical Characterisation Protocol was approved by the World Health Organization Ethics Review Committee (RPC571 and RPC572 on 25 April 2013). Institutional approval was additionally obtained by participating sites including the South Central Oxford C Research Ethics Committee in England (Ref 13/SC/0149) and the Scotland A Research Ethics Committee (Ref 20/SS/0028) for the United Kingdom, representing the majority of the data. Other institutional and national approvals were obtained by participating sites as per local requirements. Regionally appropriate decisions regarding a waiver or requirement of patient consent were made by each committee and implemented at the sites.

3. Thank you for stating the following in the Acknowledgments/ Funding Section of your manuscript:

“This work was made possible by the UK Foreign, Commonwealth and Development Office

and Wellcome [215091/Z/18/Z, 222410/Z/21/Z, 225288/Z/22/Z and 220757/Z/20/Z]; the Bill

& Melinda Gates Foundation [OPP1209135]; the philanthropic support of the donors to the

University of Oxford’s COVID-19 Research Response Fund (0009109); CIHR Coronavirus

Rapid Research Funding Opportunity OV2170359 and the coordination in Canada by

Sunnybrook Research Institute; endorsement of the Irish Critical Care- Clinical Trials Group,

co-ordination in Ireland by the Irish Critical Care- Clinical Trials Network at University

College Dublin and funding by the Health Research Board of Ireland [CTN-2014-12]; the

COVID clinical management team, AIIMS, Rishikesh, India; Cambridge NIHR Biomedical

Research Centre; the dedication and hard work of the Groote Schuur Hospital Covid ICU

Team and supported by the Groote Schuur nursing and University of Cape Town registrar

bodies coordinated by the Division of Critical Care at the University of Cape Town; the

Liverpool School of Tropical Medicine and the University of Oxford; the dedication and hard

work of the Norwegian SARS-CoV-2 study team and the Research Council of Norway grant

no 312780, and a philanthropic donation from Vivaldi Invest A/S owned by Jon Stephenson

von Tetzchner; Imperial NIHR Biomedical Research Centre; the Comprehensive Local

Research Networks (CLRNs) of which PJMO is an NIHR Senior Investigator

(NIHR201385); Innovative Medicines Initiative Joint Undertaking under Grant Agreement

No. 115523 COMBACTE, resources of which are composed of financial contribution from

the European Union’s Seventh Framework Programme (FP7/2007- 2013) and EFPIA

companies, in-kind contribution; Stiftungsfonds zur Förderung der Bekämpfung der

Tuberkulose und anderer Lungenkrankheiten of the City of Vienna, Project Number:

APCOV22BGM; Italian Ministry of Health “Fondi Ricerca corrente–L1P6” to IRCCS

Ospedale Sacro Cuore–Don Calabria; Australian Department of Health grant (3273191);

Gender Equity Strategic Fund at University of Queensland, Artificial Intelligence for

Pandemics (A14PAN) at University of Queensland, the Australian Research Council Centre

of Excellence for Engineered Quantum Systems (EQUS, CE170100009), the Prince Charles

Hospital Foundation, Australia; Brazil, National Council for Scientific and Technological

Development Scholarship number 303953/2018- 7; the Firland Foundation, Shoreline,

Washington, USA; the French COVID cohort (NCT04262921) is sponsored by INSERM

and is funded by the REACTing (REsearch & ACtion emergING infectious diseases)

consortium and by a grant of the French Ministry of Health (PHRC n°20-0424); a grant from

foundation Bevordering Onderzoek Franciscus; the South Eastern Norway Health Authority

and the Research Council of Norway; Institute for Clinical Research (ICR), National

Institutes of Health (NIH) supported by the Ministry of Health Malaysia; preparedness work

conducted by the Short Period Incidence Study of Severe Acute Respiratory Infection.

This work uses data provided by patients and collected by the NHS as part of their care and

support #DataSavesLives. The data used for this research were obtained from ISARIC4C. We

are extremely grateful to the 2648 frontline NHS clinical and research staff and volunteer

medical students who collected these data in challenging circumstances; and the generosity of

the patients and their families for their individual contributions in these difficult times. The

COVID-19 Clinical Information Network (CO-CIN) data was collated by ISARIC4C

Investigators. Data and Material provision was supported by grants from: the National

Institute for Health Research (NIHR; award CO-CIN-01), the Medical Research Council

(MRC; grant MC_PC_19059), and by the NIHR Health Protection Research Unit (HPRU) in

Emerging and Zoonotic Infections at University of Liverpool in partnership with Public

Health England (PHE), (award 200907), NIHR HPRU in Respiratory Infections at Imperial

College London with PHE (award 200927), Liverpool Experimental Cancer Medicine Centre

(grant C18616/A25153), NIHR Biomedical Research Centre at Imperial College London

(award ISBRC-1215-20013), and NIHR Clinical Research Network providing infrastructure

support. We also acknowledge the support of Jeremy J Farrar and Nahoko Shindo.”

“Recipient: Funding received by the International Severe Acute Respiratory and emerging Infection Consortium (ISARIC).

This work was made possible by the UK Foreign, Commonwealth and Development Office and Wellcome [215091/Z/18/Z, 222410/Z/21/Z, 225288/Z/22/Z and 220757/Z/20/Z]; the Bill

& Melinda Gates Foundation [OPP1209135];

URL:

https://wellcome.org/

https://www.gov.uk/government/organisations/foreign-commonwealth-development-office

The funders had no role in study design, data collection and analysis, decision to publish, or preparation of the manuscript”

Response: Thank you. We have now deleted references to funding in the acknowledgements section. Our revised funding statement is as below:

“This work was made possible by the UK Foreign, Commonwealth and Development Office and Wellcome [215091/Z/18/Z, 222410/Z/21/Z, 225288/Z/22/Z]; and the Bill & Melinda Gates Foundation [OPP1209135]. The funders had no role in the design, analysis, manuscript preparation or decision to submit for publication. “

Response: Thank you. Apologies for the lack of clarity. The minimal dataset underlying the study results are already available within the manuscript. 

For any additional data, we have expanded on the process for seeking access under the ‘Data Availability Statement’ and is reproduced below:

“The minimal dataset underlying the results are available within the manuscript and supplementary files. Any additional data that relate to this analysis are highly detailed clinical data on individuals hospitalised with COVID-19. Due to the sensitive nature of these data and the associated privacy concerns, they are available via a governed data access mechanism following review of a data access committee. Data can be requested via the IDDO COVID-19 Data Sharing Platform (http://www.iddo.org/covid-19). The Data Access Application, Terms of Access and details of the Data Access Committee are available on the website. Briefly, the requirements for access are a request from a qualified researcher working with a legal entity who have a health and/or research remit; a scientifically valid reason for data access which adheres to appropriate ethical principles. The full terms are at https://www.iddo.org/document/covid-19-data-access-guidelines. A small subset of sites who contributed data to this analysis have not agreed to pooled data sharing as above. In the case of requiring access to these data, please contact the corresponding author in the first instance who will look to facilitate access.”

Response: Thank you. We have expanded on this in response to this previous question.

6. "One of the noted authors is a group or consortium [ISARIC Clinical Characterisation Group]. In addition to naming the author group, please list the individual authors and affiliations within this group in the acknowledgments section of your manuscript. Please also indicate clearly a lead author for this group along with a contact email address."

Response: Thank you. The lead author for the group is Laura Merson (Email address: laura.merson@ndm.ox.ac.uk). The full list of authors with affiliations is attached as a csv file with details (supplementary material). Providing the affiliations for each author in the acknowledgements would lead to a substantial increase in the number of pages (over 20 pages in a word document). We have taken this approach of listing only the author names in a previous PLoS publication without any objections (https://journals.plos.org/plosmedicine/article?id=10.1371/journal.pmed.1003969#abstract2). Please let us know if this is okay. 

Response: Thank you. We have checked and the reference list is complete and correct. 

Reviewer 1:

Well done! Very interesting findings that can contribute to more effective way in managing COVID-19 patients. There are some formatting and standard English errors that need to be reviewed. Some of the information can be misleading and confusing. The conclusion needs to be more impactful.

Response: Thank you for the encouraging and positive feedback. We have reviewed the manuscript in detail and corrected any formatting/English errors. 

We have edited the conclusions. However, given the nature of the analysis (retrospective secondary analysis of observational information from a dataset) and the limitations of our work, we have framed our conclusions cautiously. 

The revised conclusion is as below:

“Liver enyzme abnormalities are common among COVID-19 patients and associated with worse outcomes. Multiple mechanisms may explain the extent and severity of liver injury in COVID-19. Future research should focus on understanding these mechanisms, the impact of changes over time, and whether antivirals improve or worsen liver injury.”

---

## [Editor Report · Decision Letter 1]

13 Aug 2023

Liver injury in hospitalized patients with COVID-19: an international observational cohort study

PONE-D-22-30411R1

Dear Dr. Tirupakuzhi Vijayaraghavan,

We’re pleased to inform you that your manuscript has been judged scientifically suitable for publication and will be formally accepted for publication once it meets all outstanding technical requirements.

Kind regards,

Aleksandar R. Zivkovic

Academic Editor

PLOS ONE

---

## [Editor Report · Acceptance letter]

4 Sep 2023

PONE-D-22-30411R1 

Liver injury in hospitalized patients with COVID-19: an International observational cohort study 

Dear Dr. Tirupakuzhi Vijayaraghavan:

I'm pleased to inform you that your manuscript has been deemed suitable for publication in PLOS ONE. Congratulations! Your manuscript is now with our production department. 

Kind regards, 

on behalf of

Dr. Aleksandar R. Zivkovic 

Academic Editor

PLOS ONE